# FhMYB108 Regulates the Expression of Linalool Synthase Gene in *Freesia hybrida* and *Arabidopsis*

**DOI:** 10.3390/biology13080556

**Published:** 2024-07-23

**Authors:** Zhongzhou Yang, Wei Jin, Qi Luo, Xiaoli Li, Yunmin Wei, Yunlong Lin

**Affiliations:** 1Key Laboratory of Plant Genetics and Molecular Breeding, Zhoukou Normal University, Zhoukou 466001, China; yangzz121@126.com (Z.Y.); jinw877@126.com (W.J.); roach07@163.com (Q.L.); xiaoli890107@163.com (X.L.); 2College of Life Sciences and Oceanography, Shenzhen University, Shenzhen 518060, China; 3Chongqing Precision Medical Industry Technology Research Institute, Chongqing 400000, China

**Keywords:** volatile organic compound, terpenoids, *Freesia hybrida*, terpene synthase, transcriptional regulation

## Abstract

**Simple Summary:**

In this study, we screened a MYB transcription factor controlling the synthesis of linanol in *Freesia hybrida*. This transcription factor is functionally conservative and can play a role in regulating the expression of the linanol synthase gene in the model plant *Arabidopsis*, but the interaction with another bHLH type transcription factor eventually leads to different regulatory effects of the MYB regulator in *Freesia hybrida* and *Arabidopsis*.

**Abstract:**

Acting as the most abundant and widely distributed volatile secondary metabolites in plants, terpenoids play crucial roles in diverse physiological regulations and metabolic processes. Terpene synthases play a decisive role in determining the composition and diversity of terpenoids. Though the regulation of terpene synthases has been extensively investigated across various plant species, limited studies have focused on the upstream transcriptional regulation of terpene synthases. In this study, we have identified linalool as the predominant volatile compound that is released gradually from *Freesia hybrida* flowers throughout flower blooming. In the context of the transcriptome, a typical MYB transcription factor, FhMYB108, was screened based on homologous gene comparison. *FhMYB108* is capable of regulating the expression of *FhTPS1*, and both their expression levels showed gradual increase during flower opening. Moreover, *FhMYB108* exerts a stimulatory effect on the transcription of *Arabidopsis thaliana AtTPS14*, while no significant increase in *AtTPS14* expression is observed upon the stabilization of *FhMYB108* in *A. thaliana*. The highly expressed *AtMYC2* in *A. thaliana* could interact with *FhMYB108* to suppress the activation of *AtTPS14* by *FhMYB108*. The present study not only elucidates the regulatory mechanism underlying linalool synthesis but also discovers the synergistic effect of MYB and bHLH transcription factors in governing the biosynthesis of volatile terpenoids.

## 1. Introduction

Floral fragrances are secondary metabolites that are gradually secreted by odor glands in natural plants, characterized by low molecular weight, polarity, boiling point, and high fat solubility [1]. Floral fragrance, acting as a blend of naturally synthesized compounds from multiple plants in their principal habitats, exhibits significant biological functions including disease and pest resistance, as well as the attraction of insect pollinators [2]. Specific floral constituents, such as linalool, possess potent aromatic properties and find extensive applications in various fields including cosmetics, perfumery, culinary spices, and food additives [3,4,5,6]. Concurrently, a diverse array of floral constituents, such as linalool and geraniol, exhibit a broad spectrum of biological activities encompassing antibacterial, anti-inflammatory, anticancer, and antidepressant properties among others. Consequently, their application is widely prevalent in the pharmaceutical field [7,8,9,10]. Currently, over 1700 plant components have been identified as constituents of floral fragrances, primarily categorized into terpenoids, benzene rings/phenylpropanoids, and aliphatic derivatives [11,12]. Among these substances, terpenoids emerge as the predominant constituents of floral volatiles [13].

The biosynthesis of terpenoids is intricately regulated by plants throughout their growth and development, circadian rhythm, and tissue specificity [14,15,16,17,18]. The synthesis of these compounds occurs in specific plant tissues and their release into the environment is tightly regulated during precise developmental or growth stages. For example, certain plant species exhibit nocturnal flowers that emit abundant volatiles, thereby adapting to the behavior of nocturnal insects; this specific synthesis and release mechanism facilitates the enhanced adaptation of the plant to its ecological niche [19]. Terpenoid synthases, strategically positioned at the key junctures of terpenoid synthesis metabolism, possess the enzymatic capability to catalyze the conversion of FPP or GPP into a diverse repertoire encompassing monoterpenes, sesquiterpenes, and other associated compounds [20,21]. The synthesis and release of terpenoids often exhibit a positive correlation with the transcriptional levels of their corresponding terpenoid synthase genes [22]. The predominant volatile compound in *A. thaliana* is caryophyllene, accounting for approximately 90% of the total volatiles. To date, a total of 32 terpenoid synthase genes have been successfully cloned from *A. thaliana*, with *TPS11* and *TPS21* exhibiting significant expression levels and widespread involvement in the biosynthesis of caryophyllene and specific sesquiterpenes. Similarly, the production of specific volatile monoterpenes can also be attributed to several individuals, including TPS2, TPS3, TPS14, and TPS23 [23]. The release of linalool and nerolidol exhibits a positive correlation with the expression of corresponding *TPS* genes in *Snapdragons* [24]. Additionally, the *HcTPS2* gene in white ginger exhibits exclusive and specific expression solely within floral organs [25]. Therefore, it can be inferred that terpene synthase plays a pivotal role in the biosynthesis of terpenoids and serves as a crucial determinant of floral composition.

Currently, the transcriptional regulation of plant volatile terpenoids involves a repertoire of eight distinct types of transcription factors, including AP2/ERF, WRKY, NAC, bZIP, SRS, SBP, MYB, and bHLH family member [26]. In *A. thaliana*, AtMYB21 and AtMYB24 exhibit the positive regulation of flower development, wherein the expression levels of *AtTPS11* and *AtTPS21* gradually increase during flower development to facilitate the release of sesquiterpene caryophyllene [27,28,29]. The transcription factor AtMYC2 plays a pivotal role in the development of flower stamens and demonstrates specific recognition ability towards the promoters of *AtTPS11* and *AtTPS21* genes, thereby directly regulating their expression and facilitating the release of sesquiterpene caryophyllene [30,31]. However, the synthesis and transcriptional regulation of terpenoids constitute a highly intricate process that has thus far been exclusively investigated in *A. thaliana*; nevertheless, the precise underlying mechanism remains elusive. Moreover, the investigation of transcription factors derived from flowers, which play a crucial role in volatile emission, is currently lacking in the scientific literature. Therefore, it is crucial to identify a species exhibiting a robust floral fragrance in order to conduct comprehensive investigations aimed at elucidating the transcriptional regulatory mechanisms governing volatile terpenoid production in plant flowers.

*Freesia hybrida*, a bulbous herbaceous flower belonging to the iris family, exhibits opulent blooms and exquisite floral esthetics, rendering it an ideal specimen for investigating the intricate realm of botanical fragrance and floral metabolism [32,33]. The identification and evaluation of aromatic volatile compounds in 26 cultivated and 8 hybrid *Freesia hybrida* revealed that terpenoids, particularly monoterpenoids, constituted the predominant class of VOCs across most germplasm. Notably, linalool and d-limonene exhibited the highest levels. Additionally, hybrid varieties exhibit significant advantages in terms of flavor and VOC accumulation across cultivated species, thereby providing valuable insights for the evaluation of genetic resources, floral aroma breeding, and the advancement of commercial applications in *Freesia hybrida* [34]. In “Shiny Gold”, a *Freesia hybrida* variety, the main volatile components are also terpenoids such as linalool, β-ocimene, and d-limonene, and multiple *TPS* genes seem to be involved in the synthesis of volatile compounds and the release of floral fragrance [35]. In this study, FhMYB108, a transcription factor involved in the regulation of terpenoid synthase gene expression, was identified through screening the *Red River*^®^ *Freesia hybrida* transcription database. Furthermore, the molecular mechanism underlying the regulatory role of *FhMYB108* in volatile terpenoid synthesis was further elucidated. This study establishes a theoretical framework to elucidate the molecular mechanism underlying plant floral development, thereby providing a robust foundation for enhancing plant and flower traits.

## 2. Materials and Methods

### 2.1. Plant Materials and Growth Conditions

The *F. hybrida* cultivar *Red River*^®^ was cultivated under a controlled greenhouse at a temperature range of 20 °C–25 °C, with a photoperiod of 16 h light and 8 h darkness. The flowers were collected from different individuals at various stages of flowering for the measurement of volatiles, extraction of RNA, and gene expression analysis.

*A. thaliana* (Columbia-0) was used in this study. Following a vernalization treatment at 4° C for a period of 2–3 days, the specimen was then transferred to soil and grown under controlled conditions at room temperature, maintaining a photoperiod of 16 h light followed by an 8 h dark phase. The preparation of *A. thaliana* protoplasts involved the utilization of fresh and tender leaves cultivated for a duration of 3–4 weeks, while transgenic functional validation required *A. thaliana* seedlings grown for a period of 6–8 weeks.

### 2.2. Volatiles Collection and Analysis

The volatiles within *F. hybrida* were collected by using solid-phase headspace microextraction. Briefly, the fresh flowers were meticulously arranged at various developmental stages within a hermetically sealed glass enclosure, while the polydimethylsiloxane (PDMS) solid-phase microextraction needle was carefully inserted 2 cm above the flower through the rubber pad positioned on top of the glass cover. The entire apparatus was positioned within a controlled environment chamber, simulating an artificial climate incubator, where the volatile compounds emitted by *F. hybrida* flowers were collected at a temperature of 25 °C for 2 h. Then, the extracted needles were subsequently subjected to a gas chromatography-mass spectrometry (GC-MS) analysis, and the resulting mass spectrum was compared with the standard spectrum (NIST2008) for the qualitative identification of volatile components.

### 2.3. Quantitative Real-Time PCR

Total RNA was isolated with TRIzol reagents. First-strand cDNA was synthesized from 1 μg of total RNA using the RevertAid First Strand cDNA Synthesis Kit (cat. no. K1621; Thermo Fisher, Waltham, MA, USA). Then, qPCR (qRT-PCR) was carried out by SYBR Green. All the primers used for qRT-PCR were designed by using the National Center for Biotechnology Information (NCBI) online software Primer-BLAST (https://www.ncbi.nlm.nih.gov/tools/primer-blast/index.cgi?LINK_LOC=BlastHome) (Appendix A), and the 18S rRNA was used as internal controls to normalize gene expression.

### 2.4. Sequence Analysis and Phylogenetic Analysis

The *A. thaliana* AtMYB21 sequence was utilized as a bait for conducting BLAST analysis to identify potential MYB transcription factors involved in the transcriptional regulation of terpene synthases in *F. hybrida*. Then, the candidate genes were amplified using specific primers. The identification of homologous proteins across diverse organisms was accomplished by employing the BLAST algorithm available on NCBI (https://blast.ncbi.nlm.nih.gov). Phylogenetic analysis was carried out with the program MEGA5 using the neighbor-joining method and the phylogenetic tree was then displayed using the iTOL online service (https://itol.embl.de/). Sequence alignment was generated by the MEGA5 software and then was visualized using DNAMAN.

### 2.5. Subcellular Localization and BiFC Assays

For the subcellular localization assay, the genes were cloned into the HA-pUC19-GFP (p35S: GFP) vector, and these recombinant plasmids were subsequently introduced into the protoplasts of *A. thaliana* via PEG-mediated transformation as previously reported [36]. For the BiFC assay, the genes were cloned into the HA-pUC19–GFPN vector carrying N-terminal GFP and HA-pUC19-GFPC vector containing C-terminal GFP, respectively, and these recombinant plasmids were also introduced into the protoplasts of *A. thaliana* s with PEG-mediated transformation. Fluorescence in subcellular localization and BiFC assays was observed using a confocal fluorescence microscope at 20 hpi.

### 2.6. Transcriptional Activity Analysis

The transcriptional activation activity of FhMYB108 was evaluated by constructing a transient transformation vector with a GD label, which was driven by the 35S promoter. The concentrated pUC19-GD (Negative control), pUC19-GD-FhMYB108, and pUC19-GD-VP16 (positive control) plasmids were co-transfected into *A. thaliana* protoplasts along with Gal4: GUS, respectively. The GUS activity was assessed following incubation at 22 °C in darkness for 21–23 h. In order to investigate the activation of a specific gene by the transcription factor FhMYB108, we cloned the gene promoter into the HA-pUC19: GUS vector and co-transformed it with pUC19-FhMYB108 as mentioned above.

### 2.7. Overexpression of Exogenous Genes in Arabidopsis thaliana

To further substantiate the regulatory role of FhMYB108 in linalool metabolism, a plant transformation overexpression vector was constructed and subsequently introduced into wild-type *A. thaliana* (Col) via the Agrobacterium-mediated pollen tube channel method. In brief, the *FhMYB108* sequence was amplified using cDNA as a template, seamlessly cloned and ligated into the pBI121 vector, and subsequently transformed into *Agrobacterium*. We suspended *Agrobacterium* in a beaker containing 5% sucrose solution supplemented with surfactants. Next, we immersed the flowers of mature, non-podded wild-type *A. thaliana* in a beaker, ensuring their complete submersion below the liquid surface for a duration of 3 min. After labeling the plants and placing them in a water-filled tray, they were shielded from light overnight before being subjected to the normal conditions of cultivation the following day.

### 2.8. Statistical Analysis

The GraphPad Prism was used to conduct the statistical analysis. The mean ± standard deviation is reported based on the results obtained from a minimum of three independent experiments. Student’s *t*-test was employed to assess differences between the groups, with a significance level of *p* < 0.05 indicating statistical significance.

## 3. Results

### 3.1. Composition and Release Regulations of Volatile Compounds in F. hybrida

The blossoming of flowers is accompanied by the release of a plethora of volatile compounds, functioning as chemical cues to attract insects for the purpose of pollination. Here, the GC-MS analysis revealed that the volatile compounds emitted during the blooming phase of *Red River^®^* flowers were predominantly composed of 18 monoterpenes, 5 sesquiterpenoids, and 3 carotenoid derivatives (Table 1). Among them, the monoterpenes constituted 82.41% of the content, whereas carotenoid derivatives accounted for 16.84%, and sesquiterpenes contributed a mere 0.75% (Figure 1A). The results of further qualitative analysis revealed that the volatile compounds identified in the sample were primarily composed of linalool (44.83%), followed by terpineol (22.88%). Additionally, a certain quantity of limonene (D-limonene) (5.05%), ionone (β-Ionone) (12.66%), dihydroionone (dihydro-πIonone) (3.15%), myrcene (2.66%), and basil (E)-ocimene) were detected (Figure 1B).

By the monitoring of volatile compounds emission at various stages of flower blooming in *F. hybrida*, it becomes evident that the quantity of released volatile substances gradually increases as fragrant orchid flowers unfold (Figure 1C). The levels of linalool and terpineol release gradually increase in parallel with the floral blooming, reaching their peak during the stage of maximum bloom (the green bud stage (S1) and the red–green interphase stage (S2), the red bud stage (S3), early opening stage (S4), and full opening stage (S5)) (Figure 1D). Moreover, the release of other terpenoids was also consistent with this principle (Table 2). The findings demonstrate the unequivocal dominance of linalool among the volatile compounds present in *F. hybrida*, thereby emphasizing its significant biological implications for this species.

### 3.2. Relevant Regulatory Molecules of Linalool

Currently, a total of eight terpene synthase genes (*FhTPS1 FhTPS2 FhTPS3 FhTPS4 FhTPS5 FhTPS6 FhTPS7,* and *FhTPS8*) have been identified in *F. hybrida* [37]. Among them, it has been discovered that *FhTPS1* exclusively utilizes GPP as a substrate to generate linalool, which potentially serves as the primary source of the volatile compounds in the *F. hybrida* flowers. To quantify the expression of *FhTPS1* across diverse developmental stages and floral organs, including petals, pistils, sepals, receptacles, and stamens, our findings revealed a robust expression pattern of *FhTPS1* in petals, pistils, and stamens; conversely, its expression was significantly lower in sepals and receptacles (Figure 2A). The crucial aspect lies in the striking similarity between its expression level and the release pattern of linalool, which demonstrated a gradual increase that aligns with flower blooming (Figure 2B). This observation strongly suggests that the catalytic activity of FhTPS1 may be solely accountable for linalool in the *F. hybrida* flowers.

Based on previous transcriptome data [3], we identified three full-length cDNA sequences encoding MYB transcription factors in *F. hybrida*, namely *FhMYB21L1* (591 bp), *FhMYB21L2* (591 bp), and *FhMYB108* (789 bp). These sequences were obtained through homologous comparison with the *AtMYB21* and *AtMYB24* transcription factors in *A. thaliana*, known for their regulatory role in terpenoid metabolism. The regulatory functions of *FhMYB21L1* and *FhMYB21L2* in terpenoid metabolism have been experimentally validated, while the role of *FhMYB108* remains unreported [38]. The FhMYB108 exhibits two characteristic R2R3-MYB transcription factor domains, renowned for their ability to specifically bind to MYB elements (Figure 2C). Additionally, the phylogenetic analysis reveals that FhMYB108 and other MYB-type transcription factors implicated in the regulation of terpenoid metabolism form a distinct cluster, indicating a potential role for FhMYB108 in governing terpenoid metabolic processes (Figure 2D).

The expression patterns of *FhMYB108* were found to be downregulated during the S1 and S2 stages while exhibiting upregulated during the S3, S4, and S5 stages of flower development (Figure 2E). The regular expression pattern is consistent with both the expression pattern of *FhTPS1* and the release pattern of linalool, suggesting that *FhMYB108* potentially serves as a regulator for *FhTPS1* expression. Moreover, *FhMYB108* exhibits negligible expression levels in petals, pistils, and receptacles; however, it demonstrates prominent expression specifically in stamens (Figure 2F).

### 3.3. FhMYB108 Is a Transcription Factor with Strong Activation Activity

The transient expression vector harboring *FhMYB108* fused with a GFP tag was constructed and subsequently transfected into protoplasts. Intense green fluorescence emanated from the nucleus upon fluorescence, indicating the nuclear localization of FhMYB108 similar to that of the red NLS nuclear localization signal marker. These findings suggest that FhMYB108 functions as a nuclear-localized transcription factor (Figure 3A).

GD can interact with Gal4 and subsequently modulate *GUS* gene expression through GD-associated proteins. In this study, the plasmids pUC19-GD (negative control), pUC19-GD-FhMYB108, and PUC19-GD-VP16 (positive control) harboring the GD label were co-transfected with Gal4: GUS, respectively, followed by the subsequent evaluation of GUS activity. The robust transcriptional activation potential of FhMYB108 is evident, highlighting its role as a proficient transcriptional activator (Figure 3B).

Additionally, the *TPS1* gene promoter was cloned to generate a HA-pUC19-FhProTPS1: GUS plasmid, which was co-transfected with the transient expression vector HA-pUC19-*FhMYB108* into *A. thaliana* protoplasts (HA-pUC19-CAT serving as a negative control). The result suggests that FhMYB108 has the ability to significantly enhance GUS activity, indicating its autonomous binding capability to the promoter region of *FhTPS1* and potential for transcriptional activation (Figure 3C). With truncation analysis, it has been determined that FhMYB108 is likely to exert regulatory control over the expression of *FhTPS1* by exhibiting a strong binding affinity towards the −1016~−916 bp region of the *FhTPS1* promoter (Figure 3D).

### 3.4. The Activation of Sesquiterpene and Monoterpene Synthase Genes by FhMYB108 in Arabidopsis thaliana

Due to the unavailability of a reliable genetic transformation system in *F. hybrida*, we conducted a transient transformation of FhMYB108 in *A. thaliana* protoplasts to validate its regulatory role in terpenoid metabolism. FhMYB108 demonstrates a close evolutionary relationship with MYB21, thereby suggesting its potential regulatory role in modulating the expression of *A. thaliana* sesquiterpene synthase genes *AtTPS11* and *AtTPS21*. Additionally, linalool biosynthesis is catalyzed by *AtTPS14* in *A. thaliana*. Therefore, we cloned the promoter sequences of *AtTPS11*, *AtTPS14*, and *AtTPS21* and subsequently assessed the activation potential of FhMYB108 on these promoters (with HA-pUC19-CAT serving as a negative control) in accordance with the previous experiments on transcription factor activation assays. The results demonstrated that the co-transfection of FhMYB108 with the promoters of *AtTPS14* and *AtTPS21* resulted in a significant increase in the GUS values, thereby indicating the specific binding ability of FhMYB108 to the promoter regions of AtTPS14 and AtTPS21, consequently activating their transcription (Figure 4A).

When FhMYB108 was transiently expressed solely in protoplasts, we observed a pronounced upregulation of only *AtTPS14*, responsible for linalool biosynthesis, compared to the control group. However, there were no notable changes detected in the expression of *AtTPS21* (Figure 4B).

### 3.5. The Expression of TPS14 and Release of Linalool in FhMYB108 Overexpressing Seedlings

The regulatory role of FhMYB108 was further elucidated by successful expression in *A. thaliana* through inflorescence transformation. The mRNA extraction was conducted to quantify the expression level of *AtTPS14*; however, no significant upregulation in its expression level was observed upon overexpressing *FhMYB108* (Figure 5A). Additionally, the GC-MS analysis consistently revealed a negligible linalool content with no discernible alterations observed in the transgenic line when compared to the wild type (Figure 5B). Based on the transient expression results, we propose the presence of a putative transcription factor in *A. thaliana* that potentially interacts with FhMYB108 or competes for binding to the promoter region of *AtTPS14*, thereby exerting robust inhibitory effects on *AtTPS14* gene expression.

### 3.6. The Expression of the AtTPS14 Gene by FhMYB108 and AtMYC2 in Arabidopsis

In *A. thaliana*, in addition to MYB21 and MYB24, which have been experimentally shown to modulate terpene metabolism, an additional bHLH-type transcription factor (AtMYC2) has also been identified as a regulator of terpene biosynthesis through the jasmonic acid (JA) pathway based on research reports [39]. The transcription factor AtMYC2 plays a pivotal role in the regulation of sesquiterpenoid biosynthesis, exhibiting high expression levels specifically in *Arabidopsis* flowers and demonstrating reported interactions with MYB21. The results reported herein demonstrate that *Arabidopsis thaliana* flower volatiles primarily comprise sesquiterpene caryophyllene (>90%) with minimal levels of linalool, suggesting that AtMYC2 may not play a significant role in promoting linalool accumulation. Based on these results, we can make the following bold assumptions: (1) AtMYC2 is incapable of activating the expression of linalool synthase gene AtTPS14; (2) In *Arabidopsis thaliana*, it is plausible that AtMYC2 interacts with FhMYB108 to suppress the activation of FhMYB108 on AtTPS14. The absence of any discernible impact on linalool content resulting from *FhMYB108* overexpression can plausibly be attributed to the involvement of AtMYC2. By conducting transcription factor activation experiments, we observed a strong binding affinity of AtMYC2 towards the promoters of *TPS11* and *TPS21*, while no discernible binding capability was demonstrated towards the promoters of *TPS14* (Figure 6A). In contrast, the transient expression of *AtMYC2* in protoplasts resulted in a significant upregulation of *TPS11* and *TPS21*, while no alteration was observed for the *AtTPS14* gene, suggesting that AtMYC2 lacks the ability to positively regulate linalool synthase gene expression (Figure 6B).

Additionally, the interaction between FhMYB108 and AtMYC2 was validated through a double fluorescence complementation assay in this study. The results demonstrated that the fluorescence signal was exclusively observed upon the co-transfection of FHMYB108-GFPN and ATMYC2-GFPC, thereby indicating a potential interaction between FhMYB108 and AtMYC2. Furthermore, FhMYB108 and AtMYC2 were individually or co-transferred with At_Pro_*TPS14*: GUS into the protoplasts of *A. thaliana* leaves to quantify alterations in binding affinity. The findings illustrate a pronounced transcriptional activation by FhMYB108 on the *TPS14* promoter, while AtMYC2 exhibits an inability to activate the transcription of the *TPS14* promoter. Moreover, the co-expression of AtMYC2 and FhMYB108 significantly diminishes the transcriptional activation capacity of FhMYB108.

## 4. Discussion

To optimize their chances of survival and reproductive success, plants commonly emit a substantial quantity of volatile small molecule compounds, particularly terpenoids, as chemical signaling agents to establish connections among plants or between plants and animals. These compounds play pivotal roles in essential processes such as pollination, the induction of ovulation, and even direct or indirect involvement in plant defense [40,41,42]. Therefore, elucidating the composition of plant volatile terpenoids is crucial for comprehending the physiological functions and potential practical applications.

The release of terpenoids exhibits spatiotemporal specificity, with peak emission occurring during the period of full flower bloom or fruit ripening. The *Red River*^®^ cultivar of *F. hybrida* at different stages of flower opening is selected as the experimental material in this study to investigate the expression pattern of the terpenoid synthase gene (*FhTPS1*), which plays a crucial role in regulating the biosynthesis of its major volatile compound (linalool). The results revealed a gradual upsurge in the expression of FhTPS1 in *F. hybrida* during flower blooming, exhibiting a similar trend as observed in *Clarkia Breweri* [43]. The presence of this regular expression suggests the existence of a regulatory system in *F. hybrida* that governs the expression of the linalool synthase gene, indicating an intricate mechanism for controlling secondary metabolism in this plant species.

Currently, there is a lack of a systematic approach in the investigation of transcriptional regulation mechanisms governing terpenoid biosynthesis. On one hand, the existing studies lack coherence and fail to elucidate the synergistic relationships among multiple transcription factors in a comprehensive manner. On the other hand, the majority of investigations have predominantly focused on specific cash crops such as tobacco and tomato while overlooking more significant flower plants. Among the numerous transcription factors reported, only AtMYB21 and AtMYC2 exhibit dual regulatory functions in both terpenoid metabolism and development, specifically in stamen formation, in *A. thaliana*. Previous investigations on anthocyanins and proanthocyanidins in *F. hybrida* have underscored the potential significance of MYB transcription factors in secondary metabolism, thereby prompting our focus on elucidating the role of MYB transcription factors in linalool synthesis and regulation.

By employing AtMYB21 as a bait for comparative transcriptome analysis in *F. hybrida*, we successfully identified FhMYB108, a transcription factor with the potential to regulate terpenoid metabolism. The phylogenetic grouping of FhMYB108 and AtMYB21 suggests potential functional similarities between them. Furthermore, experimental evidence derived from assays on transcription factor activation has substantiated the specific binding capability of FhMYB108 to the promoter regions of *AtTPS11* and *AtTPS21* (which are known targets of AtMYB21), thereby effectively inducing their transcriptional activity. Meanwhile, FhMYB108 exhibits a comparable expression pattern to *FhTPS1* and demonstrates specific binding affinity towards the −916–1016 region of the *TPS1* promoter. Furthermore, FhMYB108 exhibits its regulatory role in the biosynthesis of monoterpene by activating the *TPS14* gene encoding monoterpene synthase in *A. thaliana*, providing additional evidence.

The results of promoter activation and transient overexpression demonstrate the capacity of FhMYB108 to induce the expression of linalool synthase genes in both *F. hybrida* and *A. thaliana*. However, this phenomenon was not observed upon stable expression in *A. thaliana*. The transcriptional activity of *AtTPS11* and *AtTPS21* was significantly enhanced by FhMYB108, whereas no noticeable increase in gene expression or metabolite levels was observed in Arabidopsis plants overexpressing *FhMYB108*. Through comprehensive analysis, it was determined that AtMYC2, a highly expressed transcription factor identified in both the leaves and flowers of *A. thaliana*, played a pivotal role in driving these observed changes. Notably, this transcription factor not only plays a crucial role in the terpenoid metabolism of *A. thaliana* but also exerts regulatory control over its growth and development. In stable lines, the highly expressed AtMYC2 interacts with the FhMYB108 transcription factor, thereby suppressing *TPS14* activation by FhMYB108 and subsequently inhibiting linalool biosynthesis and other related metabolites.

## 5. Conclusions

The present study successfully cloned a MYB-type transcription factor with regulatory function in terpenoid metabolism from *Red River*^®^. The regulatory mechanism governing linalool synthesis was subsequently validated in both *F. hybrida* and *A. thaliana*. Simultaneously, it was revealed that AtMYC2 could interact with FhMYB108 and exerted a negative regulatory role in linalool synthesis. The present study elucidates the synergistic impact of MYB and bHLH transcription factors on terpenoid regulation, thereby establishing a fundamental basis for further enhancement of plant terpenoid metabolism regulatory networks.

## Figures and Tables

**Figure 1 biology-13-00556-f001:**
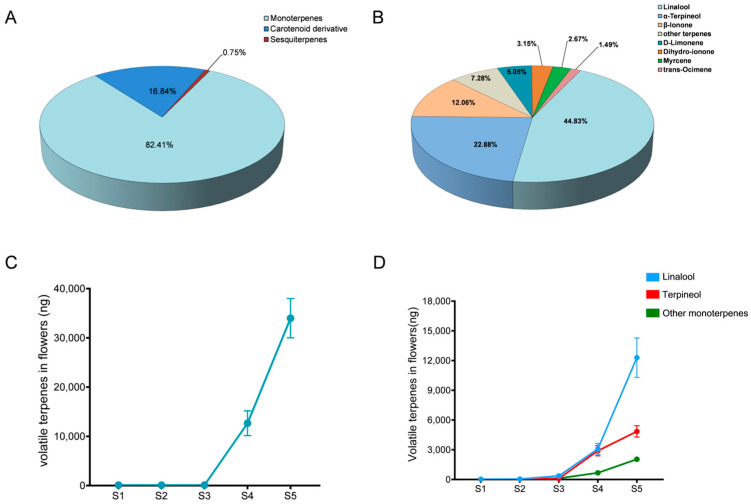
Diverse volatile terpenoids and corresponding volatilization regulations in *F. hybrida.* (**A**) The relative proportions of monoterpenes, sesquiterpenes, and carotenoid derivatives in the volatile compounds emitted by the *Red River*^®^ flowers. (**B**) The proportion of the main volatile monomer substance in the *Red River*^®^ flowers. (**C**) The release levels of volatiles in different blooming stages of *Red River*^®^. (**D**) The release levels of linalool, terpineol, and other monoterpenes in different flowering stages. In (**C**,**D**), data are shown as the mean ± SD of three biological replicates.

**Figure 2 biology-13-00556-f002:**
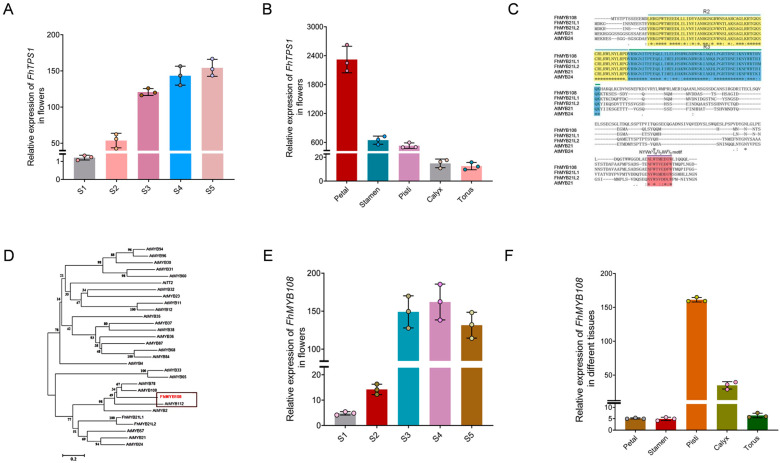
Analysis of the differential expression patterns of *FhTPS1* and *FhMYB108* in floral tissues. (**A**) The expression level of *FhTPS1* in different tissues. (**B**) The expression level of *FhTPS1* in different blooming stages of *Red River^®^*. (**C**) Amino acid sequence alignment diagram between FhMYB108 and its homologous proteins. (**D**) The phylogenetic analysis of FhMYB108. (**E**,**F**) The spatial and temporal expression pattern of *FhMYB108* in *Red River^®^* flowers. In (**A**,**B**,**E**,**F**), data are shown as the mean ± SD of three biological replicates.

**Figure 3 biology-13-00556-f003:**
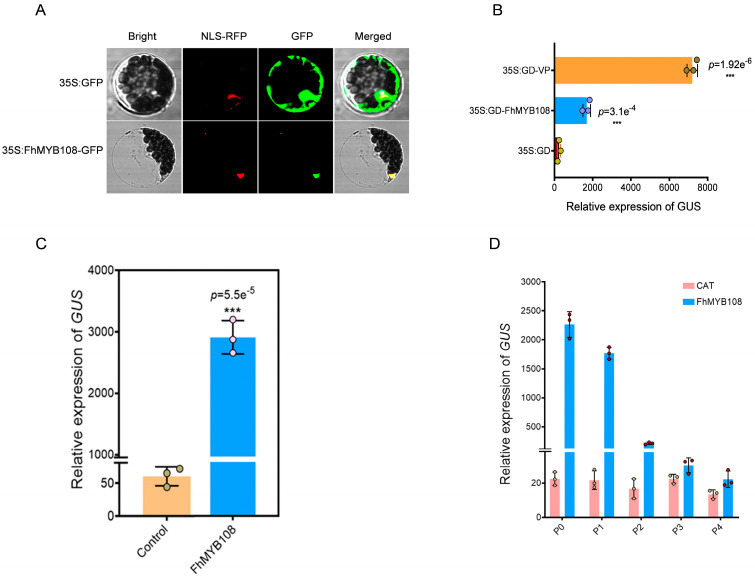
Transcriptional activity analysis of FhMYB108. (**A**) The localization of FhMYB108 in cells. (**B**) The measurement of *GUS* reporter gene expression. (**C**) The binding assay of FhMYB108 to *TPS1* promoter. (**D**) The activation validation of FhMYB108 to the truncated promoter of *TPS1*. In (**B**,**C**,**D**), data are shown as the mean ± SD of three biological replicates. Differences were as-sessed using a two-tailed *t*-test (***, *p* < 0.001) in (**B**,**C**).

**Figure 4 biology-13-00556-f004:**
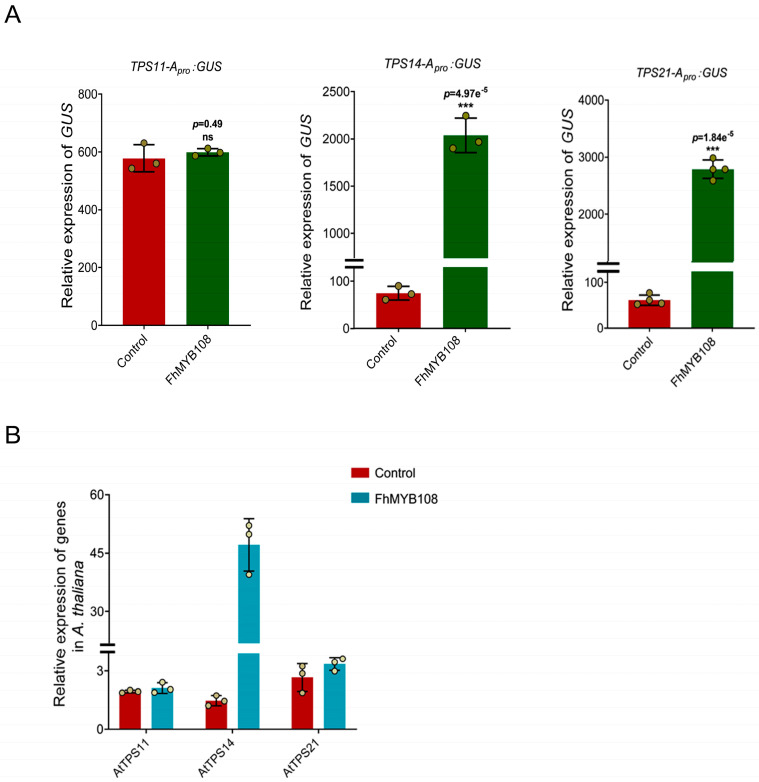
Regulation of gene expression by the transient expression of *FhMYB108* in *A. thaliana*. (**A**) Promoter activation assay in *A. thaliana*. (**B**) The expression level of terpene synthase genes in *FhMYB108* overexpressed *A. thaliana*. In (**A**,**B**), data are shown as the mean ± SD of three biological replicates. Differences were assessed using a two-tailed *t*-test (***, *p* < 0.001; ns, *p* > 0.05) in (**A**).

**Figure 5 biology-13-00556-f005:**
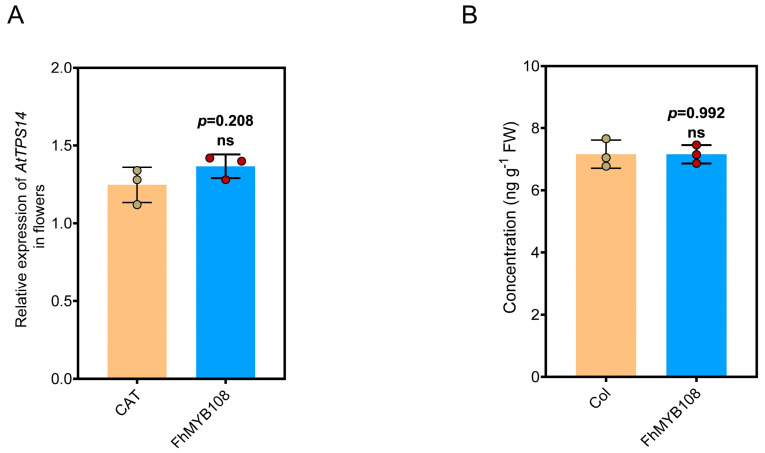
*TPS14* expression and linalool release in *FhMYB108* overexpressed *A. thaliana*. (**A**) The quantification of the expression level of *AtTPS14* upon the overexpression of *FhMYB108* in *A. thaliana*. (**B**) Linalool content in overexpressed seedlings. In (**A**,**B**), data are shown as the mean ± SD of three biological replicates. Differences were assessed using a two-tailed *t*-test (ns, *p* > 0.05) in (**A**).

**Figure 6 biology-13-00556-f006:**
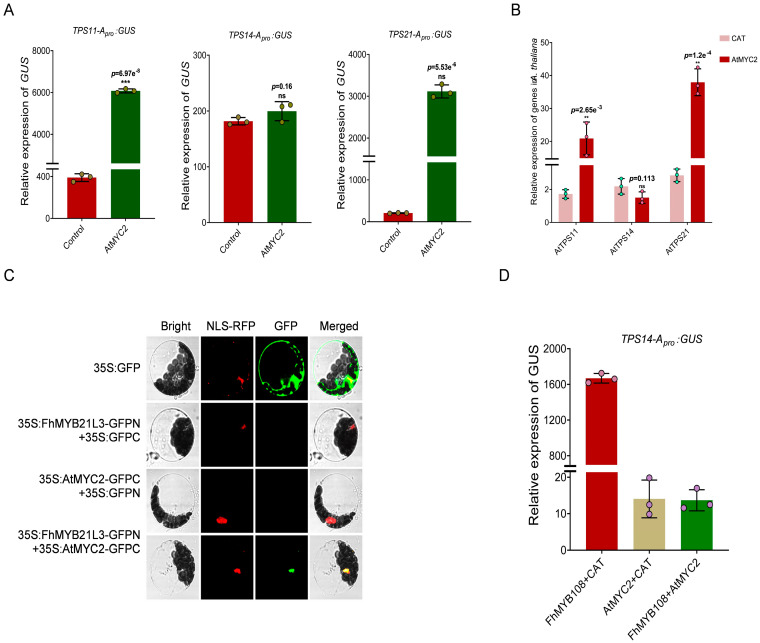
Interaction between FhMYB108 and AtMYC2. (**A**) The transcriptional activation assay of AtMYC2. (**B**) The expression of terpene synthase genes in AtMYC2 overexpressed *A. thaliana*. (**C**) The interaction between FhMYB108 and AtMYC2 verified by BiFC. (**D**) FhMYB108 and AtMYC2 jointly regulate the transcription of *AtTPS14*. In (**A**,**B**,**D**), data are shown as the mean ± SD of three biological replicates. Differences were assessed using a two-tailed *t*-test (***, *p* < 0.001; **, *p* < 0.01; ns, *p* > 0.05) in (**A**,**B**).

**Table 1 biology-13-00556-t001:** Qualitative and quantitative analysis of volatile terpenes in *Red River*^®^ flowers.

Categories	RT (min)	Compounds	Formula	Relative Contents
Monoterpenes	5.036	α-Pinene	C_10_H_16_	0.25%
	6.089	Sabinene	C_10_H_16_	0.42%
	6.182	β-pinene	C_10_H_16_	0.15%
	6.476	Dehydrocineole	C_10_H_16_O	0.06%
	6.662	Myrcene	C_10_H_16_	2.67%
	7.413	α-Terpinene	C_10_H_16_	0.89%
	7.707	Eucalyptol	C_10_H_18_O	0.69%
	7.865	D-Limonene	C_10_H_16_	5.05%
	8.138	trans-Ocimene	C_10_H_16_	1.49%
	8.458	Terpinene	C_10_H_16_	0.27%
	8.985	cis-Linaloloxide	C_10_H_18_O_2_	0.28%
	9.418	Furanoid linalool oxide	C_10_H_18_O_2_	0.24%
	9.629	Terpinolene	C_10_H_16_	0.39%
	9.922	Linalool	C_10_H_18_O	44.83%
	12.028	4-Terpineol	C_10_H_18_O	0.32%
	12.407	α-Terpineol	C_10_H_18_O	22.88%
	12.726	Decanal	C_10_H_20_O	0.31%
	12.872	β-Cyclocitral	C_10_H_16_O	0.38%
Sesquiterpenes	16.077	Cycloisosativene	C_15_H_24_	0.16%
	16.212	Copaene	C_15_H_24_	0.04%
	17.163	Farnesene	C_15_H_24_	0.11%
	17.627	g-Gurjunene	C_15_H_24_	0.19%
	18.867	Nerodilol	C_15_H_26_O	0.24%
Carotenoid derivatives	16.875	Dihydro-ionone	C_13_H_22_O	3.15%
	17.525	β-Ionone	C_13_H_20_O	12.06%
	16.699	Ionone	C_13_H_20_O	0.14%

**Table 2 biology-13-00556-t002:** Release of volatile compounds from *Red River*^®^ flowers at different stages of blooming (ng g^−1^ FW).

Compounds	S1	S2	S3	S4	S5
α-Pinene	n.d	n.d	n.d	23.58 ± 2.16	30.87 ± 4.52
β-pinene	n.d	n.d	n.d	40.99 ± 4.85	24.86 ± 3.10
Dehydrocineole	n.d	n.d	n.d	n.d	20.83 ± 1.48
Myrcene	n.d	n.d	n.d	59.02 ± 4.87	72.65 ± 4.88
α-Terpinene	n.d	n.d	n.d	6.89 ± 2.01	38.15 ± 3.75
D-Limonene	n.d	n.d	15.89 ± 4.02	42.89 ± 4.17	155.21 ± 8.55
trans-Ocimene	n.d	n.d	48.56 ± 3.41	120.36 ± 10.25	635.1 ± 26.55
Cis-ocimene	n.d	n.d	24.26 ± 3.88	189.25 ± 10.52	745.12 ± 20.78
cis-Linaloloxide	n.d	n.d	n.d	70.68 ± 7.45	147.29 ± 18.26
Terpinolene	n.d	n.d	n.d	8.88 ± 6.02	46.21 ± 5.12
Linalool	n.d	29.32 ± 2.06	158.36 ± 7.69	6845.48 ± 204.36	17,963.14 ± 364.88
4-Terpineol	n.d	n.d	n.d	24.84 ± 2.86	34.29 ± 5.14
α-Terpineol	8.26 ± 2.06	18.26 ± 3.07	135.98 ± 8.66	6889.21 ± 59.64	8795.66 ± 276.11
Cycloisosativene	n.d	n.d	n.d	10.25 ± 1.87	33.26 ± 3.55
γ-Gurjunene	n.d	n.d	n.d	22.02 ± 4.20	73.58 ± 2.55
Nerodilol	n.d	n.d	n.d	21.55 ± 3.47	37.98 ± 5.64
Dihydro-ionone	n.d	n.d	3.02 ± 2.55	588.21 ± 60.54	908.23 ± 40.88
β-Ionone	n.d	n.d	8.22 ± 4.20	2701.84 ± 105.32	4429.14 ± 207.44
Ionone	n.d	n.d	n.d	n.d	65.88 ± 6.45

(The data are represented by means ± standard deviations (*n* = 3). n.d: not detected).

## Data Availability

Data are contained within the article or Appendix A.

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
