# Peer review of "FhMYB108 Regulates the Expression of Linalool Synthase Gene in Freesia hybrida and Arabidopsis"

_biology, 2024, doi:10.3390/biology13080556_

Round 1
Reviewer 1 Report
Comments and Suggestions for Authors
The aim of the study is to explore a predominant volatile compound in Freesia hybrida: linalool. Results are significant and demonstrate that the Tf FhMYB108 regulates the expression of a linalool synthase gene, the FhTPS1 gene. Some changes are suggested. In the ‘Introduction’ section: • The description of the state of the art about the treated research topic is insufficient. I suggest to insert recent works, discuss them, and to underline the value added of the submitted work. For instance, the following works could be cited: https://www.mdpi.com/1420-3049/26/15/4482, https://www.mdpi.com/2223-7747/9/11/1597 In the ‘Results’ section: • In Table 2, the stages of blooming are not explained. Their means are present below, at Lines 151-152 • A more concise and structured organization of the results achieved could improve the quality of the work Some minor issues: • In the manuscript some gene do not appear in the Italic form – for instance, at Lines 17, 383, 393 • The scientific name of Arabidopsis thaliana must be written in full – Line 21 • Please check the references format. https://www.mdpi.com/journal/biology/instructions
Comments on the Quality of English LanguageMinor editing of English language required
Author Response
Response to Reviewer 1’s comments
Overall Comment: The aim of the study is to explore a predominant volatile compound in Freesia hybrida: linalool. Results are significant and demonstrate that the Tf FhMYB108 regulates the expression of a linalool synthase gene, the FhTPS1 gene. Some changes are suggested.
Response: Thank you very much for your appreciation and valuable comments of our manuscript and these comments provide important guidance to our researches. We have revised our manuscript carefully to make it more suitable to be published.
Comment 1: In the ‘Introduction’ section: • The description of the state of the art about the treated research topic is insufficient. I suggest to insert recent works, discuss them, and to underline the value added of the submitted work. For instance, the following works could be cited: https://www.mdpi.com/1420-3049/26/15/4482, https://www.mdpi.com/2223-7747/9/11/1597.
Response: Thanks for your comment and reminding. Several recent literatures focusing on the selection of varieties and analysis of aromatic volatile compounds in Freesia hybrid had being incorporated in the introduction section. The literature is as follows:1. Weng, S.; Fu, X.; Gao, Y.; Liu, T.; Sun, Y.; Tang, D. Identification and evaluation of aromatic volatile compounds in 26 cultivars and 8 hybrids of Freesia hybrida. Molecules 2021, 26, 4482. https://doi.org/10.3390/molecules26154482.2. Srinivasan, A.; Ahn, M.S.; Jo, G.S.; Suh, J.N.; Seo, K.H.; Kim, W.H.; Kang, Y.I.; Lee, Y.R.; Choi, Y.J. Analysis of Relative Scent Intensity, Volatile Compounds and Gene Expression in Freesia “Shiny Gold”. Plants 2020, 9, 1597. https://doi.org/10.3390/plants9111597
Comment 2: In Table 2, the stages of blooming are not explained. Their means are present below, at Lines 151-152.
Response: Thanks for your comment and reminding. We have transferred the description of the stages of blooming from “Results section 2.2” to “Results section 2.1”. Thanks again for your valuable comments.
Comment 3: A more concise and structured organization of the results achieved could improve the quality of the work.
Response: Thanks for your comment and reminding. We have made modifications in description of some results to make them more concise and structured organization. Thanks again for constructive comments.
Comment 4: Some minor issues: • In the manuscript some gene do not appear in the Italic form – for instance, at Lines 17, 383, 393.
Response: Thanks for your comment and reminding. We have corrected the error.
Comment 5: The scientific name of Arabidopsis thaliana must be written in full– Line 21.
Response: Thanks for your comment and reminding. For the scientific name of Arabidopsis thaliana that first appeared, we corrected it with the full name.
Comment 6: Please check the references format.
Response: Thanks for your comment and reminding. We have made modifications on some references format in accordance with the guidelines.
Reviewer 2 Report
Comments and Suggestions for Authors
In the manuscript named “FhMYB108 regulates the expression of linalool synthase gene in Freesia hybrida and Arabidopsis”, Zhongzhou Yang et al have screened a TF gene, FhMYB108, which is capable of regulating the expression of terpene synthase FhTPS1, and their expression levels exhibit a gradual increase concomitant with flower opening. Their results were interesting to readers, and their findings were also helpful for plant genetic breeding works in future, especially in terpenoid metabolism process. The manuscript was well prepared, but there were some comments about it.
(1) The function of FhMYB108 was not well documented, how authors screened this gene, please carefully explain. As authors described, authors have performed RNA-seq, this gene would response to flower development process, please add some text about it.
(2) Why used 18S rRNA gene as ref gene in RT-qPCR? Many plants researches have adopted actin or GAPDH, why 18S rRNA in present.
(3) Authors have employed AtMYB21 as a bait to blast F. hybrida genes, why not used domain analysis for identifying MYB genes in F. hybrida.
(4) Authors have described with “MEGA542” in line 369, it would be a mistake. In addition, the MEGA5 is an oldest version of MEGA, please update it with recent version.
(5) “4.7. Overexpression” line 392 is not suitable for using in section title, please add more information.
(6) Many figures have inappropriate axis, such as figure 3C, etc, please use truncature axis. In addition, the figure 6A has missed legends, please check them.
(7) Many RT-qPCR have inflated differences, such as figure 6A, which would be out of instrument detection range, please upload origin files in supplements.
Author Response
Response to Reviewer 2’s comments
Overall Comment: In the manuscript named “FhMYB108 regulates the expression of linalool synthase gene in Freesia hybrida and Arabidopsis”, Zhongzhou Yang et al have screened a TF gene, FhMYB108, which is capable of regulating the expression of terpene synthase FhTPS1, and their expression levels exhibit a gradual increase concomitant with flower opening. Their results were interesting to readers, and their findings were also helpful for plant genetic breeding works in future, especially in terpenoid metabolism process. The manuscript was well prepared, but there were some comments about it.
Response: Thank you very much for your appreciation and valuable comments of our manuscript and these comments provide important guidance to our researches. We have tried our best to revise our manuscript according to your comments and the corresponding responses to your specific comments are as follows.
Comment 1: The function of FhMYB108 was not well documented, how authors screened this gene, please carefully explain. As authors described, authors have performed RNA-seq, this gene would response to flower development process, please add some text about it.
Response: Thanks for your comment and reminding. Utilizing previous transcriptome data as the reference library [3], we employed AtMYB21 and AtMYB24, known to be implicated in linalool synthesis in Arabidopsis thaliana, as bait for blast analysis, resulting in the identification of three homologous genes. Among them, FhMYB108 was selected as the core point of our investigation in view of the consistent expression pattern with TPS1.3. Yang, Z., Li, Y., Gao, F., Jin, W., Li, S., Kimani, S., Yang, S., Bao, T., Gao, X., & Wang, L. MYB21 interacts with MYC2 to control the expression of terpene synthase genes in flowers of Freesia hybrida and Arabidopsis thaliana. J Exp Bot 2020, 71, 4140-4158. https://doi.org/10.1093/jxb/eraa184.
Comment 2: Why used 18S rRNA gene as ref gene in RT-qPCR? Many plants researches have adopted actin or GAPDH, why 18S rRNA in present.
Response: Thanks for your comment and reminding. Actin or GAPDH are commonly employed as internal reference genes in various plant species; however, due to variations in species and tissues, 18s is frequently utilized as a stable quantitative internal reference gene in Freesia hybrid and it has been extensively reported [4, 5].4. Shan, X., Li, Y., Yang, S., Yang, Z., Qiu, M., Gao, R., Han, T., Meng, X., Xu, Z., Wang, L., & Gao, X. The spatio-temporal biosynthesis of floral flavonols is controlled by differential phylogenetic MYB regulators in Freesia hybrida. New Phytol 2020, 228, 1864–1879. https://doi.org/10.1111/nph.16818.5. Li, Y., Shan, X., Zhou, L., Gao, R., Yang, S., Wang, S., Wang, L., Gao, X. The R2R3-MYB factor FhMYB5 from Freesia hybrida contributes to the regulation of anthocyanin and proanthocyanidin biosynthesis. Front Plant Sci 2019, 9, 1935. https://doi.org/10.3389/fpls.2018.01935.
Comment 3: Authors have employed AtMYB21 as a bait to blast F. hybrida genes, why not used domain analysis for identifying MYB genes in F. hybrid.
Response: Thanks for your comment and reminding. AtMYB21 possesses multiple functional domains, thus comparing a blast with only one of its domains may lead to inaccuracies. Therefore, we opted for the entire sequence comparison to ensure precision and avoid errors.
Comment 4: Authors have described with “MEGA542” in line 369, it would be a mistake. In addition, the MEGA5 is an oldest version of MEGA; please update it with recent version.
Response: Thanks for your comment and reminding. We are sorry for the spelling mistake, the word “MEGA542” should be “MEGA5”, and also, we have updated it with recent version.
Comment 5: “4.7. Overexpression” line 392 is not suitable for using in section title, please add more information
Response: Thanks for your comment and reminding. We had changed the inappropriate title “Overexpression” to “Overexpression of exogenous genes in Arabidopsis thaliana”
Comment 6: Many figures have inappropriate axis, such as figure 3C, etc, please use truncature axis. In addition, the figure 6A has missed legends, please check them.
Response: Thanks for your comment and reminding. The images have been thoroughly examined and necessary modifications have been implemented to rectify the issues with the files. In Figure 6A, the horizontal axis corresponds to the legends. Therefore, we had chosen not to add additional legends.
Comment 7: Many RT-qPCR have inflated differences, such as figure 6A, which would be out of instrument detection range, please upload origin files in supplements.
Response: Thanks for your comment and reminding. Some genes share significantly different temporal and spatial expression levels, which may lead to substantial variations in multiples. We have provided the data utilized for generating the bar chart and quantitative outcomes. Kindly review them.
Round 2
Reviewer 2 Report
Comments and Suggestions for Authors
Thanks for authors works about the revision, most of comments have been well addressed, but there was still one comment remaining, many figures have inappropriate axis, please adjust them for better displaying.
Author Response
Overall Comment: Thanks for authors works about the revision, most of comments have been well addressed, but there was still one comment remaining, many figures have inappropriate axis, please adjust them for better displaying.
Response: Thank you very much for your appreciation and valuable comments of our manuscript and these comments provide important guidance to our researches. We have revised our manuscript carefully to make it more suitable to be published. The images have been thoroughly examined and necessary modifications have been implemented to rectify the issues with the files. The inappropriate axes in figures, such as figure 2, 4, and 6 have been adjusted by using truncature axes, so that all data could be visually displayed from the column graphs. We hope the modification proposed herein can align with your requirements.